# Comparison of the Effects of DOTA and NOTA Chelators on ^64^Cu-Cudotadipep and ^64^Cu-Cunotadipep for Prostate Cancer

**DOI:** 10.3390/diagnostics13162649

**Published:** 2023-08-11

**Authors:** Inki Lee, Min Hwan Kim, Kyongkyu Lee, Keumrok Oh, Hyunwoo Lim, Jae Hun Ahn, Yong Jin Lee, Gi Jeong Cheon, Dae Yoon Chi, Sang Moo Lim

**Affiliations:** 1Department of Nuclear Medicine, Korea Institute of Radiological & Medical Sciences, Seoul 01812, Republic of Korea; inkilee2080@kirams.re.kr; 2Research Institute of Radiopharmaceuticals, FutureChem Co., Ltd., Seoul 04793, Republic of Korea; minhwan.kim@futurechem.co.kr (M.H.K.); kyongkyu.lee@futurechem.co.kr (K.L.); keumrok.oh@futurechem.co.kr (K.O.); hyunwoo.lim@futurechem.co.kr (H.L.); 3Division of Applied RI, Korea Institute of Radiological & Medical Sciences, Seoul 01812, Republic of Korea; ajphd@kirams.re.kr (J.H.A.); yjlee@kirams.re.kr (Y.J.L.); 4Graduate School of Translational Medicine, Seoul National University College of Medicine, Seoul 03080, Republic of Korea; 5Department of Nuclear Medicine, Seoul National University College of Medicine, Seoul 03080, Republic of Korea; larrycheon@snu.ac.kr

**Keywords:** prostate cancer, positron emission tomography, prostate-specific membrane antigen, theranostics, copper-64, DOTA, NOTA

## Abstract

Background: This study compared the effects of 1,4,7,10-tetraazacyclododecane-1,4,7,10-tetraacetic acid (DOTA) and 1,4,7-triazacyclononane-1,4,7-triacetic acid (NOTA) as ^64^Cu-chelating agents in newly developed prostate-specific membrane antigen (PSMA) target compounds, ^64^Cu-cudotadipep and ^64^Cu-cunotadipep, on pharmacokinetics. Methods: The in vitro stability of the chelators was evaluated using human and mouse serum. In vitro PSMA-binding affinity and cell uptake were compared using human 22Rv1 cells. To evaluate specific PSMA-expressing tumor-targeting efficiency, micro-positron emission tomography (mcroPET)/computed tomography (CT) and biodistribution analysis were performed using PSMA+ PC3-PIP and PSMA− PC3-flu tumor xenografts. Results: The serum stability of DOTA- or NOTA-conjugated ^64^Cu-cudotadipep and ^64^Cu-cunotadipep was >97%. The Ki value of the NOTA derivative, cunotadipep, in the in vitro affinity binding analysis was higher (2.17 ± 0.25 nM) than that of the DOTA derivative, cudotadipep (6.75 ± 0.42 nM). The cunotadipep exhibited a higher cellular uptake (6.02 ± 0.05%/1 × 10^6^ cells) compared with the cudotadipep (2.93 ± 0.06%/1 × 10^6^ cells). In the biodistribution analysis and microPET/CT imaging, the ^64^Cu-labeled NOTA derivative, ^64^Cu-cunotadipep, demonstrated a greater tumor uptake and lower liver uptake than the DOTA derivative. Conclusions: This study indicates that the PSMA-targeted ^64^Cu-cunotadipep can be applied in clinical practice owing to its high diagnostic power for prostate cancer.

## 1. Introduction

Prostate cancer has the highest incidence among male cancers in the United States, with 164,690 new cases that were diagnosed in 2018 [1]. In Korea, in 2018, 14,857 individuals were diagnosed with prostate cancer, which has the fourth highest incidence among male cancers [2].

Surgical resection and radiation therapy are usually used to treat local prostate cancer, and androgen deprivation therapy is applied as the first-line treatment for metastatic prostate cancer. However, metastatic prostate cancer often becomes refractory to androgen-deprivation therapy, causing castration-resistant metastatic prostate cancer [3]. Prostate cancer recurrence is detected using the prostate-specific antigen (PSA), a blood-borne tumor marker. PSA levels of ≥0.2 ng/mL post-surgery or ≥2 ng/mL from the lowest (nadir) level following radiotherapy indicate biochemical recurrence [4].

Computed tomography (CT), magnetic resonance imaging (MRI), or ^99m^Tc-methylene diphosphonate bone scan can be used to confirm the presence of recurrent or metastatic lesions in bone. For bone metastasis, bone scan diagnosis has sensitivity and specificity of only 79% and 82%, respectively. Furthermore, for lymph node metastasis, both CT and MRI exhibit low sensitivities of 7–42% and 18.8–69.7% and specificities of 82–100% and 78.6–97.6%, respectively [5]. Thus, false-negative results may occur when confirming recurrent lesions via conventional imaging tests even in situations where recurrence is clinically suspected.

Currently, radiopharmaceuticals used in positron emission tomography (PET) for diagnosing prostate cancer are being developed to primarily target the prostate-specific membrane antigen (PSMA). PSMA is overexpressed in prostate cancer cells and normally expressed in healthy human salivary and lacrimal glands, duodenum, kidneys, and colon neuroendocrine cells. Compared with conventional imaging methods, PSMA PET can diagnose locally recurrent or metastatic lesions with relatively high diagnostic accuracy (sensitivity, 63–92%; specificity, 88–100%) even at low PSA levels [3]. 

PSMA-specific ligands based on glutamate–urea–lysine structure have been developed to be primarily used for the diagnosis and treatment of prostate cancer [6]. PSMA-specific ligands for PET imaging are frequently labeled with radioactive isotopes, such as ^68^Ga and ^18^F. The diagnostic radioisotope ^68^Ga exhibits a relatively short half-life of 68 min and is produced from either ^68^Ge/^68^Ga-generator or cyclotron [7]. Conversely, ^18^F-labeled PSMA compounds exhibit a relatively long half-life of 110 min and are produced in a cyclotron [8,9,10,11,12,13,14].

^64^Cu constitutes another diagnostic radioisotope. It exhibits a relatively long half-life of 12.7 h and is produced in a cyclotron; thus, using ^64^Cu has the economic advantage of not having to purchase an additional generator from an institution already equipped with a cyclotron. 

Under the name of PSMA radioligand therapy (PRLT), PSMA target compounds are effectively being used to treat metastatic and castration-resistant prostate cancer. PRLT involves the concept of performing combined diagnosis and therapy related to the same molecular target known as “theranostics” [15]. Theranostics is a suitable method for personalized precision medicine and enables the selection of relevant patients for the treatment and prediction of their treatment response and prognosis [15]. ^64^Cu-Labeled PSMA target compounds can be used as a therapeutic radiopharmaceutical via direct labeling with ^67^Cu in the same compound [16]. Although ^67^Cu has limited availability worldwide rather than widely used ^177^Lu, the effort to improve the availability of ^67^Cu is now progressing [16]. ^67^Cu (2.58 days) has a short physical half-life compared to ^177^Lu (6.7 days); thus, ^67^Cu can be more suitable for pharmacokinetics of peptide-based drugs with fast-excretion properties [17]. 

The representative chelating agents included tetraaza macrocyclic chelator, 1,4,7,10-tetraazacyclododecane-1,4,7,10-tetraacetic acid (DOTA), 1,4,7-triazacyclononane-1,4,7-triacetic acid (NOTA), 1,4,7-triazacyclononane, and 1-glutaric acid-4,7-acetic acid (NODAGA). A comparative study involving each chelating agent reported that the ^64^Cu-NODAGA PSMA ligand increased in vivo stability more than that increased by the PSMA formulation using DOTA and exhibited considerable tumor diagnosis ability, as indicated by the high tumor-to-background ratio [18]. Furthermore, a study comparing DOTA and NOTA reported that NOTA demonstrated superior diagnostic performance than DOTA for Cu-64 labeling [19].

Herein, NOTA and DOTA chelators were compared to investigate their effects on pharmacokinetic properties and tumor targeting using ^64^Cu-cunotadipep and ^64^Cu-cudotadipep. The chemical structures of ^64^Cu-cudotadipep and ^64^Cu-cunotadipep are shown in Figure 1. An in vitro evaluation of the PSMA-binding affinity and in vivo experiments, including microPET/CT and biodistribution analysis, were conducted using ^64^Cu-labeled NOTA- or DOTA-conjugated compounds—^64^Cu-cunotadipep and ^64^Cu-cudotadipep.

## 2. Materials and Methods

### 2.1. Synthesis of ^64^Cu-Cudotadipep and ^64^Cu-Cunotadipep

^64^Cu-Cudotadipep (^64^Cu-FC705) and ^64^Cu-cunotadipep (^64^Cu-FC707) were provided by FutureChem Co., Ltd. (Seoul, Republic of Korea) [20]. A brief summary of the synthesis process is as follows: ^64^CuCl_3_ aqueous solution was dried in N_2_ gas at 100 °C, 100 μg of dotadipep or notadipep was dissolved in 0.1 M ammonium acetate (0.1 mL, pH 4.5) and mixed with the dried ^64^CuCl_3_, and the mixture was then stirred at room temperature for 10 min. The reaction mixture was filtered through a 0.22 μm membrane filter, and the reaction vessel was washed with distilled water (0.8 mL) and filtered. Distilled water (15 mL) was added to the filtered reaction mixture, which was gradually passed through a C-18 Plus Sep-Pak cartridge. After removing moisture from the C-18 Plus Sep-Pak cartridge using a N_2_ gas flow for 30 s, the target compounds were gradually eluted using EtOH (2 mL), and the solvent was removed via N_2_ gas flow at room temperature to obtain ^64^Cu-cudotadipep (2.9 mCi, 96.7% RCY) or ^64^Cu-cunotadipep (2.88 mCi, 96% RCY). 

### 2.2. Stability Analysis

The serum stabilities of ^64^Cu-cudotadipep and ^64^Cu-cunotadipep were evaluated in human and mouse sera for 48 h. Briefly, 10 µL ^64^Cu-cudotadipep (3.7 MBq) or ^64^Cu-notadipep (3.7 MBq) was mixed with 90 µL human or mouse serum and then incubated at 37 °C for 48 h. At each time point (2, 24, and 48 h), a 10 µL sample was removed, mixed with 100 µL distilled water, filtered through a 0.22 µm membrane filter, and then analyzed using radio-thin layer chromatography (TLC eluent condition: 1 M NH_4_OAc:MeOH = 1:1). The radio-TLCs were performed using a radio-TLC imaging scanner (BioScan, Washington, DC, USA) for radiodetection. 

### 2.3. Cell Culture

The human prostate cancer cell line 22Rv1 was purchased from the American Type Culture Collection (Manassas, VA, USA). The human prostate cancer cell lines PC3-PIP (PSMA+) and PC3-flu (PSMA−) were kindly provided by Dr. Martin G. Pomper from Johns Hopkins University. The 22Rv1 was maintained in RPMI-1640 media containing 10% fetal bovine serum and 1% antibiotic–antimycotic solution at 37 °C and 5% CO_2_ condition. PC3-PIP and PC3-flu were maintained in RPMI-1640 media containing 10% fetal bovine serum, 1% antibiotic–antimycotic solution, and 8 µg/mL puromycin at 37 °C and 5% CO_2_ condition. 

### 2.4. Competitive Binding Assay

The 22Rv1 cells (1 × 10^6^ cells/well, 48-well plate) were washed with phosphate-buffered saline (PBS), and the culture medium was substituted with RPMI-1640 medium containing 1% bovine serum albumin. The cells were incubated for 1 h at 37 °C and 5% CO_2_ with unlabeled cudotadipep or cunotadipep (1.00 × 10^−4^–1.00 × 10^−10^ M) or 0.09–0.1 nM ^125^I-(*S*)-2-(3-((*S*)-1-carboxy-5-(3-(4-iodophenyl)ureido)pentyl)ureido)pentanedioic acid (^125^I-MIP-1095). The cells were washed twice in cold PBS and collected via centrifugation (2000 rpm, 5 min). Radioactivity was measured using 2480 WIZARD2 gamma counter (Perkin Elmer, Waltham, MA, USA). IC_50_ was calculated using Microsoft Excel and converted to Ki, using the Cheng–Prusoff equation [21].

### 2.5. Cell Uptake

PC3-PIP or PC3-flu cells (1 × 10^6^) were mixed with 3.7 MBq (100 μCi) of ^64^Cu-cudotadipep or ^64^Cu-cunotadipep and incubated at 37 °C and 5% CO_2_ for 2 h. Following incubation, the cells were washed with PBS twice and collected via centrifugation (2000 rpm, 5 min). Radioactivity was measured using 2480 WIZARD2 gamma counter (Perkin Elmer, Waltham, MA, USA). Percent (%) uptake was calculated using the ratio between the radioactivity of the cells and treated radioactivity.

### 2.6. Saturation Binding Assay

The dissociation constant (*Kd*) of MIP-1095 was determined via a saturation binding assay. The saturation binding assay was performed as performed in a previous study [8]. Briefly, the 22Rv1 cells (2 × 10^5^) were aliquoted into wells in a 24-well plate and incubated with 0.7–17.5 nM (0.0037–0.1332 MBq) of ^125^I-MIP-1095 at 37 °C for 1 h. To prevent nonspecific binding, 500 μM unlabeled MIP was added. Radioactivity was measured using 2480 WIZARD2 gamma counter (Perkin Elmer, Waltham, MA, USA). Statistical analysis was performed using Microsoft Excel.

### 2.7. Animal Experiment

All animal experiments followed the procedures approved by the Institutional Animal Care and Use Committee of Korea Institute of Radiological & Medical Sciences (approval No. KIRAMS2021-0044). Male BALB/c mice and athymic nude (Nu/Nu) BALB/c mice (weighing 20–25 g, 8 weeks old) were purchased from Nara Biotech (Seoul, Republic of Korea). PC3-PIP cells (1 × 10^7^) were dispensed in 100 μL PBS, mixed with Matrigel, and subcutaneously implanted in the right thigh of male athymic nude (Nu/Nu) BALB/c mice. When the tumor reached 0.5–0.9 cm in diameter (2–3 weeks following implantation), biodistribution studies and PET imaging were performed.

### 2.8. Biodistribution Studies

Approximately 0.93 MBq ^64^Cu-cudotadipep or ^64^Cu-cunotadipep was injected via the tail vein into normal BALB/c mice or PC3-PIP tumor-bearing nude mice (*n* = 5/group). Normal and tumor-bearing mice were euthanized at 2, 6, 24, and 48 h following injection. Tumor and normal tissues were excised and weighed, and their radioactivity was measured using a gamma counter (2480 WIZARD2 gamma counter, Perkin Elmer, Waltham, MA, USA). The radioactivity measured in each tissue was expressed as a percentage of the injected dose per gram of tissue (%ID/g).

### 2.9. MicroPET/CT Imaging

PET/CT images were obtained using a small animal PET/CT scanner (Inveon™; Siemens Preclinical Solutions, Malvern, PA, USA). Mice were anesthetized using 1.5% isoflurane and injected with 100 μL (11.10–12.95 MBq) ^64^Cu-cudotadipep or ^64^Cu-cunotadipep via the tail vein. Following radioisotope injection, PET images were taken for 25 min at 1.5, 24, and 48 h. The images were reconstructed using the three-dimensional ordered-subset expectation maximization algorithm. CT images for attenuation correction were obtained immediately following each PET scan at 50 kVp and 0.16 mA. The PET/CT images were processed using Siemens Inveon Research Workplace (IRW) software (Siemens Preclinical Solutions, Knoxville, TN, USA). The uptake values of the radioisotopes were evaluated from the region of interest of the target area, using IRW.

### 2.10. Radiation-Absorbed Dose Calculations

The biodistribution results of normal BALB/c mice were used to evaluate the human radiation-absorbed dose. The internal absorbed doses of ^64^Cu-cudotadipep and ^64^Cu-cunotadipep were evaluated using the residence time and total amount of radioactivity in the target organ of the mice. The internal absorbed dose of the mice was converted to the human dose based on the Medical Internal Radiation Dose schema, using OLINDA/EXM version 1.1 software [22,23].

### 2.11. Statistical Analysis

All data were analyzed by MS office EXCEL 2017 software. Data were described as Mean ± SD.

## 3. Results

### 3.1. Assessment of PSMA-Binding Affinity

The chemical structures of ^64^Cu-cudotadipep and ^64^Cu-cunotadipep are shown in Figure 1. The cold forms of Cu-cudotadipep and Cu-cunotadipep exhibited high binding affinity to PSMA-expressing 22Rv1 cells based on competitive binding analysis. The IC_50_ values of Cu-cudotadipep and Cu-cunotadipep were 16.84 ± 1.05 and 5.42 ± 0.64 nM, respectively. In addition, the Ki values of Cu-cudotadipep and Cu-cunotadipep that were calculated using the Bmax (0.25 nM) and *Kd* (0.13 nM) values obtained in the saturation binding test of MIP-1095 were 6.75 ± 0.42 and 2.17 ± 0.25 nM, respectively (Appendix A).

The cellular uptakes of ^64^Cu-cudotadipep and ^64^Cu-cunotadipep by PC3-PIP cells were 2.93 ± 0.06% and 6.02 ± 0.05%, respectively. The cellular uptakes of ^64^Cu-cudotadipep and ^64^Cu-cunotadipep by PC3-flu cells were 0.44 ± 0.06% and 1.52 ± 0.03%, respectively (Appendix A).

### 3.2. Serum Stability

A serum stability analysis was performed by mixing ^64^Cu-cudotadipep or ^64^Cu-cunotadipep with mouse or human serum over 2, 24, and 48 h. Both ^64^Cu-cudotadipep and ^64^Cu-cunotadipep exhibited high stabilities (over 99% and 98%, respectively) at each time point in both mouse and human sera (Appendix A).

### 3.3. Biodistribution Analysis in Normal Mice

The radioactivity of ^64^Cu-cudotadipep and ^64^Cu-cunotadipep decreased over time in each organ of normal male mice (Figure 2A,B and Table 1). ^64^Cu-Cudotadipep demonstrated a higher uptake in the liver than ^64^Cu-cunotadipep, and both compounds were confirmed to be excreted via the kidneys.

### 3.4. Biodistribution Analysis in Tumor Model

The biodistributions of ^64^Cu-cudotadipep and ^64^Cu-cunotadipep were evaluated using the PC3-PIP xenograft model (Figure 2C,D and Table 2). Both ^64^Cu-cudotadipep and ^64^Cu-cunotadipep exhibited stable uptake in the tumor for up to 48 h. At 48 h, the tumor uptake of ^64^Cu-cunootadipep was higher (28.84 ± 13.04% ID/g) than that of ^64^Cu-cudotadipep (8.62 ± 0.44% ID/g). The liver uptake of ^64^Cu-cunotadipep was approximately two-fold lower (5.74 ± 1.83% ID/g) than that of ^64^Cu-cudotadipep (13.34 ± 0.55% ID/g) at 48 h post injection.

The uptake ratio between tumor and normal tissues increased over 24 h and decreased at 48 h (Figure 3). The overall tumor-to-muscle ratio was substantially higher than the ratios between the tumor and other normal organs, including the blood, liver, and kidneys. The tumor-to-muscle ratio of ^64^Cu-cunotadipep was higher than that of ^64^Cu-cudotadipep (Figure 3 and Appendix A).

### 3.5. MicroPET/CT Imaging

PET imaging was performed on xenograft athymic nude (Nu/Nu) BALB/c mice, using PC3-PIP. Following intravenous injections of ^64^Cu-cudotadipep and ^64^Cu-cunotadipep via the tail vein, PET images were captured at 1.5, 24, and 48 h (Figure 4), and these confirmed that both ^64^Cu-cudotadipep and ^64^Cu-cunotadipep showed continuous high tumor uptake at the analyzed time points (Appendix A). The uptake of ^64^Cu-cudotadipep in the liver was relatively higher than that of ^64^Cu-cunotadipep (Appendix A).

### 3.6. Radiation Dosimetry of ^64^Cu-Cudotadipep and ^64^Cu-Cunotadipep

The estimated radiation-absorbed doses of ^64^Cu-cudotadipep and ^64^Cu-cunotadipep in each organ for adult males are shown in Appendix A. The kidneys exhibited the highest absorbed doses, 8.10 and 2.02 mGy/MBq for ^64^Cu-cudotadipep and ^64^Cu-cunotadipep, respectively.

Using tissue-weighting factors from International Commission on Radiological Protection (ICRP), the effective doses of ^64^Cu-cudotadipep and ^64^Cu-cunotadipep were measured as 3.67 × 10^−1^ and 3.00 × 10^−2^ mSv/MBq, respectively.

## 4. Discussion

In this study, we confirmed that both ^64^Cu-cudotadipep and ^64^Cu-cunotadipep, which are newly developed PSMA-targeted radiopharmaceuticals, accumulate in PSMA-expressing prostate cancer cells. Additionally, ^64^Cu-cunotadipep exhibits higher tumor uptake and lower liver uptake than those of ^64^Cu-cudotadipep.

Metastatic castration-resistant prostate cancer cells are characterized by PSMA overexpression. As a result, radiopharmaceuticals targeting overexpressed PSMA are being actively developed for PET or therapeutic applications [6,9,24]. Various PSMA-targeted ^64^Cu-labeled diagnostic agents have also been developed [25,26]. ^64^Cu-PSMA-617 has been studied in clinical trials and can be used to diagnose metastatic lesions with high resolution [24]. ^64^Cu-PSMA-617 can be effectively used for the lymph node staging of prostate cancer [27], and for diagnosing gastric and PSMA-expressing prostate cancers [26,28,29,30].

Using the same compound, it is possible to perform the theranostic application with a diagnostic (^64^Cu, ^68^Ga, etc.) and therapeutic radioisotopes (^177^Lu, ^225^Ac, etc.).

Compared with ^68^Ga, ^64^Cu has several advantages. The half-life of ^64^Cu is 12.7 h, which is longer than that of ^68^Ga (68 min) and, thus, is more suitable for obtaining delayed images [25,31]. The diagnostic use of ^64^Cu for PSMA PET imparts more sensitivity to tumor diagnosis owing to the increased tumor-to-background ratio, which enables delayed imaging the day after radiopharmaceuticals injection [25]. In addition, ^64^Cu exhibits a shorter positron range than ^68^Ga, which provides an improved spatial resolution and higher-quality images [32,33].

However, a common problem associated with ^64^Cu-labeled compounds is their high uptake in the liver and gallbladder [32]. Previous studies regarding bifunctional chelating agents for ^64^Cu labeling demonstrated that transchelation activity by superoxide dismutase in the liver causes free copper to separate from the chelating agent [33]. Therefore, to prevent the distortion of the PET images due to ^64^Cu use, it is essential to develop an appropriate chelating agent with high stability. The DOTA and NOTA chelating agents studied in this study are polyaza-macrocycle-based ones with a high affinity for copper [34]. In addition, compared with DOTA, NOTA can be labeled and stably maintained at a relatively low temperature [34].

Herein, both ^64^Cu-cudotadipep and ^64^Cu-cunotadipep exhibited high in vitro stability with no significant difference. Preliminary experiments confirmed that PSMA-positive prostate cancer cells demonstrated a specific uptake of both ^64^Cu-cudotadipep and ^64^Cu-cunotadipep, while PSMA-negative prostate cancer cells showed a relatively low uptake of both compounds. However, considering the biodistribution of the two radiopharmaceuticals, their in vivo stabilities are considered to be different. This difference is not because of the thermodynamic instability of ^64^Cu in combination with DOTA or NOTA. The macrocycle size of the conjugate labeled ^64^Cu exhibits a considerable effect on its in vivo stability [35], inducing the binding of free copper separated from the chelating agent to liver superoxide dismutase [36].

In terms of PSMA affinity, the Ki value of ^64^Cu-cunotadipep was approximately three-fold lower than that of ^64^Cu-cudotadipep, indicating that ^64^Cu-cunotadipep exhibits higher PSMA affinity. ^64^Cu-cudotadipep and ^64^Cu-cunotadipep have the same structure overall, only differing in the structure of the chelating agent binding ^64^Cu. However, they exhibit a large difference in their affinity for PSMA. This signifies that the binding of radiopharmaceuticals to the target is affected not only by specific binding to the receptor but also by nonspecific interactions [26]. The biodistribution analysis in normal organs demonstrated that ^64^Cu-cunotadipep uptake was high in the kidneys and that ^64^Cu-cudotadipep uptake was high in organs such as the liver and spleen. This difference in biodistribution is due to the characteristics of the chelating agents of NOTA and DOTA [37]. As previously reported, ^64^Cu-cudotadipep uptake in the liver was higher than that of ^64^Cu-cunotadipep [25,38,39]. These results suggest that the NOTA chelating agent exhibits a greater ability to stably bind ^64^Cu than the DOTA chelating agent.

^64^Cu-cudotadipep showed high uptake in the liver and spleen in the prostate cancer xenograft model, similar to that in the normal mouse model. Conversely, ^64^Cu-cunotadipep exhibited a high uptake in the blood pool immediately after injection and higher uptake in the kidneys than ^64^Cu-cudotadipep. Tumor uptake was three-fold higher for ^64^Cu-cunotadipep than ^64^Cu-cudotadipep, similar to the results of the in vitro experiment; the tumor uptake of 6^4^Cu-cunotadipep appeared to have increased because of the high PSMA affinity. The tumor-to-normal-organ-uptake ratio of both ^64^Cu-cudotadipep and ^64^Cu-cunotadipep was high 24 h after injection based on muscles as normal organs; i.e., tumor tissues can be easily distinguished from normal background tissues by using the 24 h PET images. In particular, the tumor-to-normal-organ-uptake ratio of ^64^Cu-cunotadipep was considerably higher than that of ^64^Cu-cudotadipep, suggesting that ^64^Cu-cunotadipep would be useful for clinical practice.

The uptake of both ^64^Cu-cunotadipep and ^64^Cu-cudotadipep was observed in the transplanted tumors on PET images, and these tumors could be clearly distinguished. Tumor uptake was maintained stably even in 24 and 48 h delayed imaging. As in the normal mouse biodistribution study, the hepatic uptake observed in the ^64^Cu-cudotadipep PET image was higher than that of ^64^Cu-cunotadipep. This can be attributed to the ratio of free copper from the DOTA chelating agent, which caused higher accumulation in the liver than the NOTA chelating agent.

The effective doses of ^64^Cu-cudotadipep and ^64^Cu-cunotadipep were 3.67 × 10^−1^ and 3.00 × 10^−2^ mSv/MBq, respectively, as measured using OLINDA/EXM software. The effective dose of ^64^Cu-PSMA-617 used in another study was 2.50 × 10^−2^ mSv/MBq. The effective doses of ^18^F-labeled PSMA compounds were 1.65 × 10^−2^, 2.20 × 10^−2^, 2.28 × 10^−2^, 9.21 × 10^−3^, and 1.28 × 10^−2^ mSv/MBq for ^18^F-DCFPyL, ^18^F-PSMA-1007, ^18^F-CTT1057, ^18^F-FSU-880, and ^18^F-PSMA-11, respectively [9].

The overall effective doses of ^64^Cu-cudotadipep and ^64^Cu-cunotadipep were similar to or somewhat higher than those of the various other PSMA compounds. Furthermore, the effective dose of ^64^Cu-cunotadipep was lower than that of ^64^Cu-cudotadipep. This suggested that ^64^Cu-cunotadipep use can have less radiation exposure than ^64^Cu-cudotadipep use.

Theranostics, which can provide personalized treatment based on diagnosis, is fast becoming a trend in nuclear medicine research [40]. The diagnosis and treatment of prostate cancer using PSMA is being actively studied as a field of theranostics and used in clinical practice [41]. The diagnostic abilities of ^64^Cu-cunotadipep and ^64^Cu-cudotadipep for PSMA-expressing tumors confirmed in this study are sufficient for using these compounds as PSMA-targeted theranostic tools. 

Our study indicates that ^64^Cu-cunotadipep would be more appropriate for use than ^64^Cu-cudotadipep to diagnose prostate cancer. However, radiation exposure to healthy organs should be considered if these compounds are used as chelating agents for therapeutic radionuclides. The analysis of the in vivo biodistribution showed that the overall effective dose of ^64^Cu-cunotadipep was lower than that of ^64^Cu-cudotadipep; therefore, the radiation exposure of healthy organs would be expected to be low during treatment with cunotadipep. However, chelating agents, including notadipep and dotadipep, show different biodistributions, and the effective doses of these compounds engendered high bioaccumulation in the liver and kidneys. Hence, pretreatment for the protection of healthy organs should be considered when using these therapeutic radioligands.

Unlike biological drugs and macromolecule drugs, one of the strengths of small-molecule drugs is that they have no side effects on immune response. The immunogenicity was not investigated in this study, but as can be seen in previously published works from the literature [42,43], the ^64^Cu-cudotadipep or ^64^Cu-cunotadipep used in this study is a small-molecule drug, so it is considered to have no side effects due to immunogenicity that occurs from its repeated use. 

## 5. Conclusions

In conclusion, we confirmed that the ^64^Cu-cunotadipep uptake in PSMA-expressing tumors was higher than that of ^64^Cu-cudotadipep. Furthermore, the uptake of ^64^Cu-cunotadipep in the liver was lower than that of ^64^Cu-cudotadipep. Based on these results, we suggest that ^64^Cu-cunotadipep can be used for diagnosing PSMA-positive prostate cancer in clinical practice.

## Figures and Tables

**Figure 1 diagnostics-13-02649-f001:**
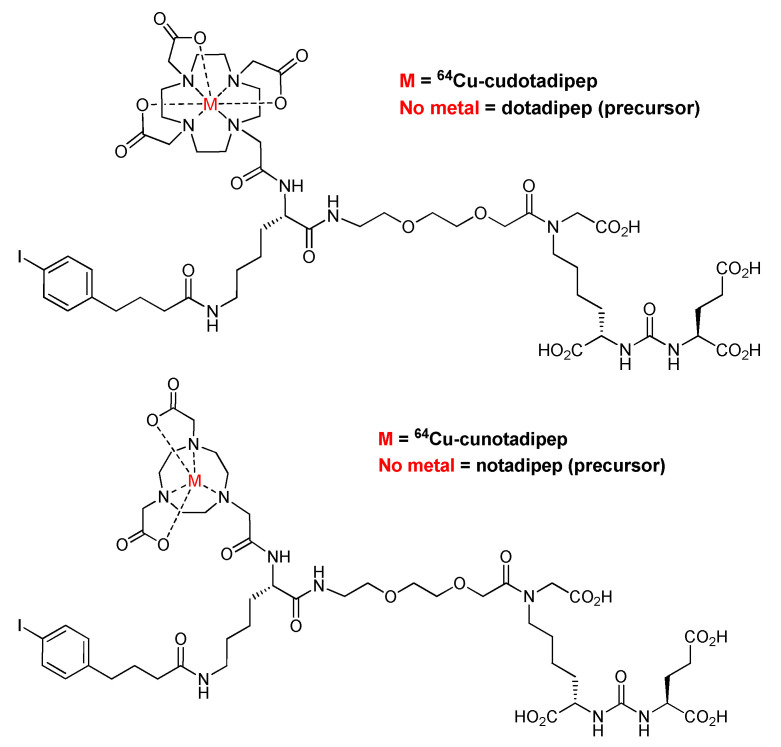
Chemical structures of ^64^Cu-cudotadipep and ^64^Cu-cunotadipep.

**Figure 2 diagnostics-13-02649-f002:**
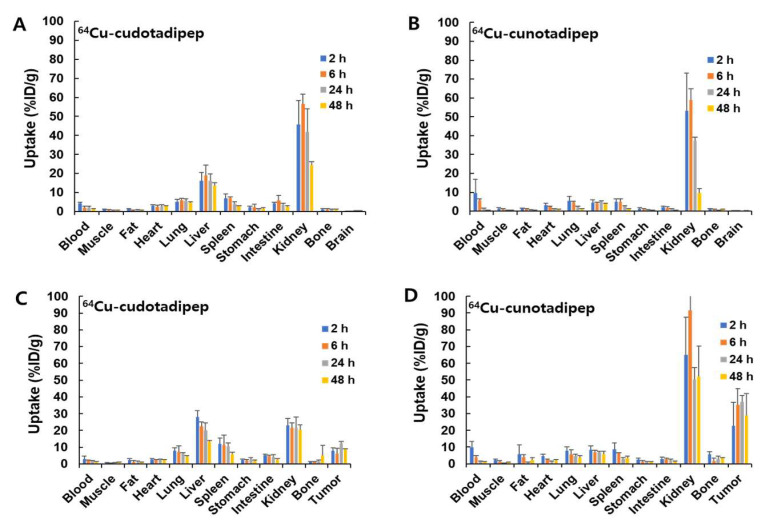
Biodistributions of ^64^Cu-cudotadipep and ^64^Cu-cunotadipep in normal BALB/c (**A**,**B**) and PC3-PIP tumor-bearing mice (**C**,**D**).

**Figure 3 diagnostics-13-02649-f003:**
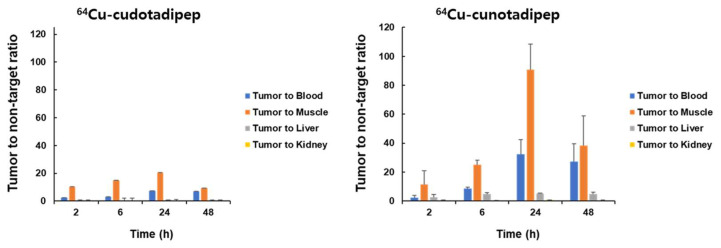
Tumor-to-non-target ratio based on biodistribution of ^64^Cu-cudotadipep and ^64^Cu-cunotadipep biodistribution in PC3-PIP tumor-bearing mice.

**Figure 4 diagnostics-13-02649-f004:**
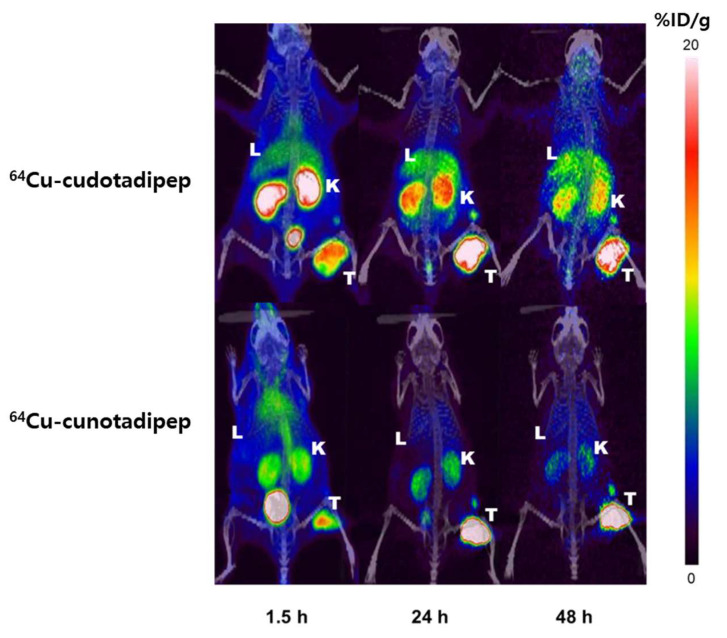
MicroPET/CT images of ^64^Cu-cudotadipep and ^64^Cu-cunotadipep. (L, liver; K, kidney; T, tumor).

**Table 1 diagnostics-13-02649-t001:** Biodistributions (%ID/g) of ^64^Cu-cudotadipep and ^64^Cu-cunotadipep in normal BALB/c mice.

	^64^Cu-Cudotadipep				^64^Cu-Cunotadipep			
	2 h	6 h	24 h	48 h	2 h	6 h	24 h	48 h
Blood	4.27 ± 0.64	2.04 ± 0.71	2.01 ± 0.92	1.22 ± 0.36	9.51 ± 7.25	5.62 ± 0.95	1.39 ± 0.15	0.45 ± 0.15
Muscle	1.01 ± 0.15	0.74 ± 0.18	0.68 ± 0.18	0.62 ± 0.12	1.30 ± 0.42	1.05 ± 0.14	0.42 ± 0.05	0.45 ± 0.04
Fat	1.26 ± 0.28	0.72 ± 0.13	0.70 ± 0.33	0.51 ± 0.27	1.20 ± 0.35	0.99 ± 0.27	0.63 ± 0.26	0.40 ± 0.10
Heart	3.02 ± 0.52	2.67 ± 0.53	2.91 ± 0.66	2.72 ± 0.39	3.17 ± 1.12	2.25 ± 0.33	1.22 ± 0.06	0.95 ± 0.07
Lung	5.16 ± 1.26	5.77 ± 1.12	5.56 ± 1.13	4.66 ± 0.50	5.48 ± 2.24	4.76 ± 0.51	2.53 ± 0.20	1.32 ± 0.13
Liver	16.12 ± 4.51	19.13 ± 5.32	16.15 ± 3.65	13.48 ± 1.56	4.38 ± 1.51	4.11 ± 0.53	5.32 ± 0.12	3.86 ± 0.31
Spleen	6.80 ± 2.46	7.09 ± 0.68	4.14 ± 0.89	2.73 ± 0.28	5.07 ± 1.38	5.05 ± 1.56	2.56 ± 0.33	1.19 ± 0.11
Stomach	2.30 ± 0.60	2.26 ± 1.59	1.38 ± 0.15	1.45 ± 0.65	1.21 ± 0.49	0.92 ± 0.45	0.67 ± 0.22	0.44 ± 0.11
Intestine	4.22 ± 0.57	5.85 ± 2.69	3.64 ± 0.83	2.53 ± 0.54	2.21 ± 0.35	1.69 ± 0.55	1.16 ± 0.24	0.56 ± 0.08
Kidney	45.74 ± 12.68	56.43 ± 5.37	41.77 ± 12.09	24.37 ± 1.88	53.28 ± 19.92	58.85 ± 6.00	37.40 ± 1.84	9.55 ± 2.33
Bone	1.09 ± 0.32	1.10 ± 0.40	0.94 ± 0.41	0.96 ± 0.25	1.07 ± 0.24	0.89 ± 0.22	0.60 ± 0.03	0.83 ± 0.10
Brain	0.26 ± 0.07	0.25 ± 0.10	0.37 ± 0.08	0.44 ± 0.13	0.26 ± 0.12	0.19 ± 0.02	0.13 ± 0.01	0.18 ± 0.01

**Table 2 diagnostics-13-02649-t002:** Biodistributions (%ID/g) of ^64^Cu-cudotadipep and ^64^Cu-cunotadipep in tumor mouse model.

	^64^Cu-Cudotadipep				^64^Cu-Cunotadipep			
	2 h	6 h	24 h	48 h	2 h	6 h	24 h	48 h
Blood	2.79 ± 1.90	1.92 ± 0.53	1.75 ± 0.41	1.28 ± 0.24	9.81 ± 3.47	4.01 ± 0.94	1.23 ± 0.38	1.08 ± 0.27
Muscle	0.78 ± 0.13	0.40 ± 0.13	0.69 ± 0.33	0.98 ± 0.24	2.14 ± 0.59	1.44 ± 0.56	0.41 ± 0.06	0.80 ± 0.22
Fat	2.22 ± 1.07	1.12 ± 0.79	1.23 ± 0.52	0.88 ± 0.42	5.72 ± 5.53	4.14 ± 1.33	0.74 ± 0.20	2.36 ± 1.25
Heart	2.68 ± 0.45	2.30 ± 0.30	2.56 ± 0.49	2.55 ± 0.12	4.56 ± 1.25	2.59 ± 0.38	1.64 ± 0.12	1.91 ± 0.57
Lung	8.02 ± 1.49	7.35 ± 3.45	6.00 ± 0.85	4.36 ± 0.62	7.72 ± 2.30	5.75 ± 2.72	4.58 ± 1.08	3.64 ± 1.14
Liver	28.09 ± 3.73	22.44 ± 2.56	20.08 ± 4.40	13.34 ± 0.55	8.33 ± 2.29	6.96 ± 1.00	6.99 ± 0.45	5.74 ± 1.83
Spleen	11.94 ± 3.50	11.43 ± 5.81	10.43 ± 1.98	5.74 ± 1.34	8.58 ± 3.99	6.63 ± 0.12	2.79 ± 0.87	3.56 ± 1.14
Stomach	2.35 ± 0.63	2.18 ± 0.47	2.66 ± 1.28	1.72 ± 0.70	2.50 ± 1.05	1.42 ± 0.53	1.27 ± 0.08	1.14 ± 0.31
Intestine	5.19 ± 0.68	4.55 ± 0.77	4.17 ± 1.45	2.56 ± 0.62	2.97 ± 1.15	2.74 ± 0.80	2.38 ± 0.42	1.54 ± 0.16
Kidney	23.11 ± 4.11	21.51 ± 3.13	21.57 ± 6.53	20.41 ± 3.03	65.02 ± 22.42	91.48 ± 12.98	50.29 ± 7.06	52.07 ± 18.32
Bone	1.34 ± 0.23	1.25 ± 0.20	1.88 ± 0.35	5.05 ± 6.09	5.71 ± 1.47	1.81 ± 1.50	2.81 ± 1.71	3.42 ± 0.40
Tumor	7.95 ± 1.58	6.03 ± 3.20	12.12 ± 1.17	8.62 ± 0.44	22.61 ± 14.10	35.14 ± 9.68	37.13 ± 3.65	28.84 ± 13.04

## Data Availability

The data presented in this study are available in this article and Appendix A.

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
