# Peer review of "Comparison of the Effects of DOTA and NOTA Chelators on 64Cu-Cudotadipep and 64Cu-Cunotadipep for Prostate Cancer"

_diagnostics, 2023, doi:10.3390/diagnostics13162649_

Round 1

Reviewer 1 Report

In this manuscript, the authors investigated the stability of the 64Cu-cudotadipep and 64Cu-cunotadipep in vitro and in vivo. The microPET/CT was used to investigate the distribution of these two materials in vivo. Based on the results, the authors suggest that the 64Cu-cunotadipep were shown to be specifically uptaken by PSMA-positive cells which could be applied in clinical practice owing to its high diagnostic ability for prostate cancer. There were still some problems in the structure and logic of this manuscript. It could be accepted if the revisions are made.

1.      The correspondence between the figure legends and the figures in this paper was confused, where are the A, B, C and D of figure 3?

2.      In the first part of the results, the authors claim that “the IC50 values of Cu-cudotadipep and Cu-cunotadipep were 16.84 ± 1.05 and 5.42 ± 0.64 nM, re-spectively”. But there are no experimental data or graphs of experimental results.

3.      It is recommended to increase experiments on the half-life of 64Cu-cudotadipep and 64Cu-cunotadipep, which is very important for the study of pharmacokinetics.

4.      There are many minor errors throughout the manuscript and the authors are advised to check the entire manuscript carefully. Such as N2 (should be N2), 1×106 (should be 1×106), 2×105 (should be 2×105), 64Cu-cudiotadipep (should be 64Cu-cudotadipep), in vitro (should be in vitro), in vivo (should be in vivo).

Moderate editing of English language required

Reviewer 2 Report

Chelators play an important role in Theranostics; the paper is relevant. The quality of the images although in the DOTA group the tumour to background ratio appears higher; what is significant is the lack of liver and splenic uptake in the NOTA-group, this is relevant.

The authors can also including that alpha therapy is also used prominently in the management of prostate cancer.

An important note in the paper is that 99mTc-MDP bone scintigraphy is no longer the preferred study of choice when evaluating Prostate cancer (Pca) metastases; 68Ga-PSMA PET/CT and other PET/CT radionuclides are now the standard of care.

Minor edits however the English and grammar are adequate.

Reviewer 3 Report

This manuscript presented by Dr. Lee and co-workers described the comparison of the pharmacokinetics between DOTA- and NOTA-based Cu-64 labeled PSMA-targeting molecules (64Cu-cunotadipep vs 64Cudotadipep). Considering the lower overall effective dose, and lower hepatic uptake, 64Cu-cunotadipep showed great advantages over 64Cu-cndotadipep.

However, the surprising kidney uptake inconsistency could be a concern and in lack of explanation.

The biodistribution of 64Cu-cunotadipep in male BALB/c mice (n=5) (Table 1) demonstrated about 50-60% ID/g at 2h and 6h, and dropped to 37% at 24h, and further dropped to less than 10% at 48h. However, in PC3-PIP tumor bearing BALB/c mice, the biodistribution of 64Cu-cunotadipep (Table 2) were over 50% at all 4 time points from 2h to 48h, and the highest was 91%. Such a high uptake in kidney, if true, significantly limited its potential in diagnostic or therapeutical applications. And its radiation exposure was higher than most of the reported PSMA compounds mentioned in the manuscript, limiting its significance and novelty.

But on the other hand, both microPET images in Figure 4 and its PET data analysis in figure S3 indicated that kidney uptake of 64Cu-cunotadipep in PC3-PIP tumor bearing BALB/c mice was lower than 8% ID/g over at 1.5h, 24h, and 48h post injection.

In addition, the biodistribution of 64Cu-cudotadipep in PC3-PIP tumor bearing BALB/c mice was around 20% ID/g, which was about half of the value of its distribution in normal BALB/c, and about half of value of 64Cu-cunotadipep in PC3-PIP tumor bearing BALB/c mice. These facts led to the conclusion that the difference of measurement methods was not the cause of significantly increased kidney uptake. 

In short, I would like to ask the author to verify the high kidney uptake in these two groups and provide an explanation for the huge difference between bio-distribution and PET imaging analysis.

Besides, the authors have cited that multiple 64Cu labeled PSMA targeting agents have been reported, eg. ref 18, 19, 27, 29, 32. A comparison of these agents in literature and the presented agents in this manuscript is strongly encouraged.

Round 2

Reviewer 1 Report

The authors have addressed all my questions, so the manuscript could be accepted now.

Author Response

Reviewer #1

Comments and Suggestions for Authors: The authors have addressed all my questions, so the manuscript could be accepted now.

à Answer: Thank you for your review and comments.

Reviewer 3 Report

Since the compound is mainly excreted via the kidneys, did the authors test if the radiopharmaceutical compound was intact in the urine. Serum stability test may not be able to address this question.

The authors should pay extra attention to the abbreviation used in the manuscript. For example, on Page 9 of 13, Line 330, "suggesting that 64Cu-cudotadipep would be useful for clinical practice." I assume it should be 64Cu-cunotadipep in this sentence. And this is one of the most significant conclusion in this manuscript. 

Again another typo on Page 10 of 13, Line 332, "The uptake of both 64Cu-cunudotadipep..."
